# Rheological Law and Mechanism for Superplastic Deformation of Ti–6Al–4V

**DOI:** 10.3390/ma12213520

**Published:** 2019-10-26

**Authors:** Chao Liu, Ge Zhou, Xin Wang, Jiajing Liu, Jianlin Li, Haoyu Zhang, Lijia Chen

**Affiliations:** School of Materials Science and Engineering, Shenyang University of Technology, Shenyang 110870, China; m18841089901@163.com (C.L.); wangxin315@163.com (X.W.); ljj951567242@163.com (J.L.); ljl651200863@163.com (J.L.); zhanghaoyu@sut.edu.cn (H.Z.); chenlijia@sut.edu.cn (L.C.)

**Keywords:** Ti–6Al–4V, activation energy of deformation, strain rate sensitivity index, grain index, hot processing map, deformation mechanism map

## Abstract

The behaviors of and mechanisms acting in Ti–6Al–4V alloy during low-temperature superplastic deformation were systematically studied by using a Gleeble-3800 thermocompression simulation machine. Focusing on the mechanical behaviors and microstructure evolution laws during low-temperature superplastic compression tests, we clarified the changing laws of the strain rate sensitivity index, activation energy of deformation, and grain index at varying strain rates and temperatures. Hot working images based on the dynamic material model and the deformation mechanism maps involving dislocation quantity were plotted on the basis of PRASAD instability criteria. The low-temperature superplastic compression-forming technique zone and the rheological instability zone of Ti–6Al–4V were analyzed by using hot processing theories. The dislocation evolution laws and deformation mechanisms of the grain size with Burgers vector compensation and the rheological stress with modulus compensation during the low-temperature superplastic compression of Ti–6Al–4V were predicted by using deformation mechanism maps.

## 1. Introduction

Ti–6Al–4V, which is characterized by formability, biocompatibility, toughness, low density, high strength, anti-corrosion, and excellent high-temperature properties, is the most widely used Ti alloy both commercially and industrially [1]. Due to its abundance and large strength-to-weight ratio, it is extensively applied in medical health, aerospace, national defense, energy, and other fields [2,3]. This alloy in equilibrium consists mainly of a hexagonal close-packed (hcp) α-phase with some body-centered cubic (bcc) β-phase at room temperature. However, the practical operations of Ti–6Al–4V are severely restricted by its superplastic deformation temperature up to 1173 K [4,5,6], which brings about problems of high tool and energy costs, low productivity, and a prolonged production cycle.

Titanium alloys, especially the rudder surface of air-to-underwater missiles that are widely used in aerospace, hardly deform at room temperature. The superplastic temperature of materials is generally above half of the melting point, and the strain rate must be within an appropriate range. The largest elongation of superplastic alloys is at least 100% and is reportedly more than 2000% in some alloys [7]. However, alloys with low formability cannot easily be manufactured into a component with a complicated shape at ambient temperature. Therefore, understanding the deformation behaviors during superplastic forming is critical for optimizing the hot forging of Ti alloys.

Superplastic deformation mechanisms are classified by three major processes. These include dominant grain boundary sliding, compatible diffusional creep, and dislocation creep [8,9]. Superplastic deformation is also accompanied by grain growth [10] and dynamic recrystallization (DRX) [11,12,13], which are related with softening. The deformation mechanism map is a new way to quantify the basis of a rate-dominating procedure. The dislocation creep map of a solid solution alloy can be illustrated by two mechanisms of dislocation viscous glide and dislocation climb. The superplastic deformation mechanism, under a certain environment, can be determined as per some workability theories, particularly deformation maps based on metallography and dislocation kinetics. These maps can be utilized fruitfully for Al alloys and composites, stainless steel, Ni and its alloy, Mg alloy, and Ti [14,15,16,17,18,19,20,21,22]. For example, a relevant map locating the major flow mechanisms in the curve of normalized stress against temperature for a certain grain size was introduced to forecast creep deformation of tungsten filament in bulbs [14]. Deformation maps of Al were drawn as functions of stress, strain size, and temperature [15,23]. The superplastic grain boundary mechanism of Fe–25Cr–20Ni austenite stainless steel was predicted by using a deformation mechanism map of grain size [16]. However, the deformation mechanism map has been rarely incorporated with the intra-grain dislocation quantity or with the dislocation breakaway solute condition mechanism in the Langdon model. Thus, new deformation mechanism maps for solid solution alloys are needed.

In this work, the low-temperature deformation of Ti–6Al–4V was investigated by using the flow curves detected from compression trials at varying tensile temperatures and strain rates. A new deformation mechanism map and a processing map of variable rheological instability concerning dislocation quantity were drawn for the two-phase Ti alloy. These were used to forecast the deformation mechanism and to quantify the dislocation quantity. This work can theoretically underlie the utilization of low-temperature deformation maps of Ti–6Al–4V [24].

## 2. Experimental

Multidirectional forging Ti–6Al–4V sheets in bar diameter of 50 mm were selected. The gauge part of each specimen was 12 mm long and 8 mm in diameter (Figure 1). The experiments were carried out on a Gleeble-3800D computer-aided servo-hydraulic analyzer (Dynamic Systems Inc., Shenyang, China) in an argon environment under homogeneous heating from 800 to 890 °C at an interval of 30 °C. The conditions were as follows: heating rate at 10 °C/s, residence time of 3 min, and strain rate at 5 × 10^−4^–0.05 s^−1^. Three samples per condition and each specimen were deformed to a real strain of 0.6 and then immediately water-cooled to ambient temperature [25,26].

In the metallographic tests, each specimen was polished mechanically, etched in 100 mL of H_2_O containing 20 mL of HF and 40 mL of HNO_3_, and then observed using a LEICA Q550IW optical microscope (LEICA Microsystems, Wetzlar, Germany). The average particle size and volume fraction of the β phase were detected on an OLYMPUS M3 image-meter (Olympus Optical Co., Tokyo, Japan). The microstructures were characterized on a Tecnai G^2^ 20 transmission electron microscope (transmission electron microscope operating conditions: TEM point resolution was 0.23 nm, line resolution was 0.14 nm, acceleration voltage was 200 kV, and magnification was 20–1,000,000) after mechanical grinding, punching, and twin-jet chemical polishing in an electrolyte of 5% HCLO_4_, 35% C_4_H_9_OH, and 60% CH_3_OH at a voltage of 20 V at 25 °C.

## 3. Results and Discussion

### 3.1. Initial Microstructure

The microstructure of the initial multidirectional forging of Ti–6Al–4V consisted of the scattered α phase (dark contrast) and the band-structured β phase (bright contrast) (Figure 2). The volume fraction of the hcp α phase is about (54 ± 3)%, and the average grain size of the bcc β phase is about (11.5 ± 0.3) μm.

### 3.2. Deformation Mechanical Behavior of Ti–6Al–4V

The deformation is nonuniform, and the deformed specimen is barrel-shaped due to the inevitable interfacial friction between the tested specimen and dies. Essentially, the computed flow stress–strain plots overestimate the real ones. For these reasons, the impact of friction on flow stress must be taken into account, which makes the stress–strain curves closer to the real ones. The details of flow stress correction have previously been reported [27,28]. The experimental real stress–strain curves of Ti–6Al–4V were all delineated within 800–890 °C and the strain rate within 0.005−0.05 s^−1^ (Figure 3a–d). The figures show an upper yield stress and sudden drop when the plastic deformation begins. The linear "initial rise" is elastic deformation, and the flow stress begins at the same point as plastic deformation. Clearly, the flow stress rose in the beginning of low-temperature deformation and then decreased or remained in a steady state (Figure 3a–d). At this point, the hardening and softening of the alloy reach an equilibrium state in the process of plastic deformation. The initial drop of upper yield stress occurred during the thermal compression deformation of this alloy at 860 or 890 °C (Figure 3c,d). This was because the rise of deformation temperature contributed to dislocation migration and grain boundary sliding and harbored the conditions for dynamic recrystallization. Therefore, the yield stress initially dropped. However, because dynamic recrystallization cannot fully proceed and softening was insignificant when the deformation temperature was lower than 900 °C and the strain rate was fast (0.05 s^−1^), the upper yield effect occurred.

The rheological strain curve under the typical thermal deformation conditions (860 °C, strain rate at 0.05 s^−1^) was selected for analysis (Figure 4). Clearly, at the starting of deformation (strain ε < 0.021), the work hardening was very significant, and its rate (θ = ∂σ/∂ε) was above 1000 MPa. The θ started to drop at ε of 0.021–0.108 and then minimized and stabilized gradually with the increment of ε beyond the critical DRX ε_c_. According to the Kupiec, M. model [29,30], the rheological stress was enhanced rapidly at the initial stage (zone I) but gradually stabilized after the strain exceeded ε_c_, which is typical of dynamic restoring. However, after the rheological stress of Ti–6Al–4V peaked to σ_p_, it quickly declined, and the difference from the rheological stress of the theoretical model was ∆σ. This suggests that the alloy was dominant in the DRX softening mechanism (zone II), and when the strain ε was close to 0.6, it started to reached the stable state (zone III). The softening effect of this alloy during high-temperature deformation fell within the strain of ε = 0.14–0.57. Based on the above method, the strain ranges of the softening effect at different strain rates to reach the alloy temperature of 860 °C have been summarized in Table 1. Clearly, the high-temperature deformation softening behaviors of this alloy were largely affected by the strain rate sensitivity (Table 1). 

Figure 5 illustrates the strain rate sensitivity (m) of the superplastic flow. As the strain rate reduced, m elevated. In particular, m exceeded 0.7 within the strain rate of 5 × 10^−4^–5 × 10^−3^ s^−1^ at 800–890 °C, indicating that the considerable elongations found above were caused by the superplastic compression behaviors [31].

Low-temperature compression tensile deformation of Ti–6Al–4V is a heated stimulation procedure. The flow stress σ can be expressed as a function of strain rate $\dot{\varepsilon }$ and deformation temperature T [32]:(1)$$\sigma =k\varepsilon^{n}{\dot{\varepsilon }}^{m}\exp\left(\frac{Q}{RT}\right)$$ where *k* and *R* are constants, *m* is the strain rate sensitivity index, *ε* is the strain rate, *n* is the hardening index (n≈0 for a superplastic material), and *Q* is the activation energy of deformation. At the constant strain rate, *Q* can be determined as follows: (2)$$Q=2.303R\times {\left[\partial \mathrm{lg}/\partial \left(1/T\right)\right]}_{\dot{\varepsilon }}\times {\left[\partial \mathrm{lg}\dot{\varepsilon }/\partial \mathrm{lg}\sigma \right]}_{T}$$ where ${\left[\partial \mathrm{lg}\dot{\varepsilon }/\partial \mathrm{lg}\sigma \right]}_{T}=\frac{1}{m}$; thus,
(3)$$Q=2.303R\times {\left[\partial \mathrm{lg}\sigma /\partial \left(1/T\right)\right]}_{\dot{\varepsilon }}\times \frac{1}{m}$$

With the data from the compression tests, the lgσ vs. 1/T was plotted and was found to have a slope of ${\left[\partial \mathrm{lg}\sigma /\partial \left(1/T\right)\right]}_{\dot{\varepsilon }}$. It was substituted together with m into Equation (3) to determine *Q* (Figure 6).

Table 2 lists the calculated activation energy *Q* of Ti–6Al–4V. With the variation of strain rate, the *Q* ranged from (328.88 to 546.9 ± 7.7) kJ/mol (Table 2). *Q* elevated with the rising strain rate, but first increased and then declined with the rise of the deformation temperature.

The activation energy of thermal deformation of Ti–6Al–4V is far larger than the activation energy for self-diffusion cited in the literature (Table 3). This was because twin-induced DRX, accompanied by severe dislocation motion, occurred during deformation. Thus, the activation energy for deformation rose significantly. 

### 3.3. Microstructures after Superplastic Compression Deformation

The microstructure evolution rules of metal materials during high-temperature deformation considerably affect the hot forming properties. The grain size was computed as follows [36]: (4)d=1.74L where *d* is average grain size; *L* is the linear intercept of metallopraphical microstructures. Figure 7 shows the metallographs of Ti–6Al–4V at 860 °C and strain rates of 5 × 10^−4^, 10^−3^, 5 × 10^−3^, and 5 × 10^−2^ s^−1^. Based on Equation (4) and Figure 7, the grain sizes of this alloy under the above deformation conditions were calculated to be 28.13, 24.43, 16.93, and 9.41 μm, respectively.

The Arrhenius constitutive relation universal model [36] was expressed as follows: (5)$$\dot{\varepsilon }=\frac{AD_{0}Gb}{kT}{\left(\frac{b}{d}\right)}^{p}{\left(\frac{\sigma }{G}\right)}^{1/m}\exp\left(-\frac{Q}{RT}\right)$$ where *d* is the grain size, *σ* is the stress, *p* is the grain index, *D_0_* is the diffusion coefficient, *b* is the Burgers vector, *G* is the shear module, *k* is the Boltzmann constant, and *R* is the ideal gas constant. 

Figure 8 shows the relation curves of strain rate—grain size of Ti–6Al–4V. Based on Equation (5), the straight-line slope is the grain index *p*, which reflects the effect of grain size on the thermoforming performance. The grain index at 860 °C is 2.31. With the above method, the *p* of this alloy at other temperatures can be determined (Figure 9). 

### 3.4. Thermal Processing Maps of Ti–6Al–4V

According to the dynamic materialogic model, the plastic history of this material can be regarded as an energy dissipative system, and the energy dissipation is decided by the material processing rheological behaviors and obeys the power-law Equation [37]: (6)$$\sigma =k\times {\dot{\varepsilon }}^{m}$$ where σ is the rheological stress; *k* and m are constants decided by the strain rate $\dot{\varepsilon }$. Based on the energy dissipation theory, Prasad et al. divided the system energy *P* during metallic thermal deformation into dissipation energy (*G*) and dissipation assisting energy (*J*). The consumed energy or dissipation energy (*G*) is mostly converted to thermal energy and slightly stored as crystal defect energy in the material. The dissipated energy, resulting from structural changes during deformation, is called the assisting dissipation energy (*J*).
(7)P=G+J
(8)$$P=\sigma \times \dot{\varepsilon }=\int_{0}^{\dot{\varepsilon }}\sigma \times d\dot{\varepsilon }+\int_{0}^{\sigma }\dot{\varepsilon }\times d\sigma$$

The allocation between *J* and *G* is decided by the material rheological properties, and the *J* to *G* ratio is called the strain rate sensitivity index (m, same as the appeal strain rate sensitivity index m value): (9)$$m=\frac{dJ}{dG}={\left(\frac{\partial \left(\log\sigma \right)}{\partial \left(\log\dot{\varepsilon }\right)}\right)}_{\varepsilon .T}$$

At given temperature (*T*) and strain (*ε*): (10)$$J=\int_{0}^{\sigma }\dot{\varepsilon }\times d\sigma =\sigma \times \dot{\varepsilon }\times m/\left(m+1\right)$$

When at *m* = 1, the material is at the ideal linear dissipation state, and *J* maximizes to *J_max_*: (11)$$J_{\max}=\frac{\sigma \times \dot{\varepsilon }}{2}$$

Based on Equations (10) and (11), the expression of power dissipation rate (η) can be deduced:(12)$$\eta =\frac{J}{J_{\max}}=\frac{2m}{m+1}$$

The Prasad instability criterion is a dynamic material model built on the basis of the large plastic deformation continuum mechanics and irreversible thermodynamics theories. Based on Ziegler’s rationale of maximum entropy production rate, Prasad et al. thought the dissipation function D ($\dot{\varepsilon }$) and the strain rate $\dot{\varepsilon }$ satisfied an inequality: (13)$$\frac{dD}{d\dot{\varepsilon }}<\frac{D}{\dot{\varepsilon }}$$

When the system is unstable and the material rheological instability criterion is [38], then the following is true: (14)$$\xi \left(\dot{\varepsilon }\right)=\frac{\partial ln\left(\frac{m}{m+1}\right)}{\partial ln\dot{\varepsilon }}+m<0$$

Based on Equations (12) and (14), we plotted the power dissipation rate and rheological instability of Ti–6Al–4V during thermal compression and thereby acquired the thermal processing maps (Figure 10). The changes of power dissipation rate can be expressed by contour curves, and the shaded areas correspond to the rheological instability zones.

The strain rate of this alloy within 800–840 °C is 0.034–0.05 s^−1^, and the thermal deformation falls within the rheological instability zone (Figure 10). The strain rates at 800–840 and 860–875 °C are 0.0005–0.001 and 0.001–0.005 s^−1^, respectively, and the power dissipation rate *η* is up to 0.45–0.5. At this moment, m is very large (0.49–0.73), which does not fall within the instability zone. Thus, Ti–6Al–4V has very high thermal forming performance at low-temperature and low-strain-rate areas and at medium-temperature and medium-strain-rate areas. 

### 3.5. Deformation Mechanism of Ti–6Al–4V with Dislocation Quantity

Severe dislocation motion occurred during the thermal deformation of Ti–6Al–4V. Therefore, based on the Ruano–Wadsworth–Sherby (RWS) deformation mechanism, we introduced the dislocation model via a constitutive equation and plotting deformation mechanism maps that involved the dislocation quantity, grain size, strain rate, and rheological stress. 

#### 3.5.1. Construction of Deformation Mechanism

The high-temperature deformation of metals can be described by a constitutive equation [39]: (15)$${\dot{\varepsilon }}_{i}=A_{i}{\left(\frac{b}{d_{i}}\right)}^{p}\times \frac{D}{K\times T\times b^{2}}\times {\left(\frac{\sigma_{i}}{E}\right)}^{n}$$ where $A_{i}$, *m*, and *p* are material constants, *E* is Young’s modulus, $\sigma_{i}$ is stress, *b* is Burgers vector, ${\dot{\varepsilon }}_{i}$ is the steady-state strain rate, and $d_{i}$ is the particle size; *D* is the diffusion index, including the lattice index *D_L_* and the crystal boundary index *D_gb_*. 

The internal dislocation root count of single crystal grains ($n_{i}$) can be computed as follows [40]: (16)$$n_{i}=2\left[\left(1-\nu \right)\times \pi \times d_{i}\times \tau_{i}\right]/\left(Gb\right)$$ where v is Poisson’s ratio, and $\tau_{i}=0.5\sigma_{i}$ is the shear stress (MPa). The compression test data of Ti–6Al–4V at 830 or 890 °C were substituted into Equations (15) and (16), and thereby, the crystal grain sizes of included dislocation count underlying the RWS deformation mechanism were plotted, with module compensationary stress as the X-axis and Burgers vector compensation as the Y-axis. The physical parameters used in the computation are listed in Table 4.

#### 3.5.2. Application of Deformation Mechanism

The RWS deformation mechanism maps involving dislocation count at 830 or 890 °C were drawn in Figure 11. The normalized grain size with Burgers vector normalization ((*d/b*) × 10^−7^) and the normalized flow stress with modulus normalization ((*σ/E*) × 10^4^) of Ti–6Al–4V at the above temperatures were calculated (Table 5). 

At 830 °C and a very high strain rate, Ti–6Al–4V fell within the (3.95 × 10^7^) (6.85 × 10^8^) (52) (71) (368) dislocation polygon zone. The mechanism was the dislocation glide controlled by lattice diffusion, with a stress index of 7 (Figure 11a and Table 5). As the strain rate dropped, it fell within the (3.95 × 10^7^) (69,425) (5425) (5816) (368) dislocation polygon zone, and the mechanism was the dislocation glide controlled by lattice diffusion with a stress index of 5. 

At 890 °C and a very high strain rate, Ti–6Al–4V fell within the (7.25 × 10^6^) (64,908) (6488) (4258) (415) dislocation polygon zone (Figure 11b and Table 5). The mechanism was the dislocation glide controlled by lattice diffusion with a stress index of 5. As the strain rate declined, the alloy deformation mechanism was unchanged. 

Based on Figure 11a,b and the thermal deformation behavioral characteristics, it was found that at 830 °C and a fast strain rate, the deformation of Ti–6Al–4V followed a lattice diffusion control mechanism with a stress index of 7. At this moment, m and *Q*-value were high, the power dissipation rate *η* was small, and it fell within the rheological instability zone. When at 890 °C, the deformation of this alloy obeyed the lattice diffusion-controlled mechanism with a stress index of 5. At this moment, m significantly rose, the *Q*-value declined, *η* was close to the peak value, and the alloy did not fall within the rheological instability zone. This was because the deformation of Ti–6Al–4V at 890 °C was favorable for and intensified the dislocation motion (Figure 12a). Figure 12a shows that large amounts of dislocations exist in α phase grains, which indicates that the dislocation movement plays an important role in superplastic deformation of the as-received alloy. Compared with Figure 12a,b, when increasing the deformation temperature, the regime controlled by superplastic grain boundary sliding significantly expanded, while that controlled by dislocation pipe grain boundary sliding decreased Lots of dislocations became seriously consumed, and large grain boundary was formed, which is conducive to grain boundary slip. This indicates that superplastic grain boundary sliding is the dominant deformation mechanism at deformation temperature. Thus, the stress index at this state was low, and the forming performance was excellent.

## 4. Conclusions

(1) The m of Ti–6Al–4V at 800–890 °C with a strain rate of 5 × 10^−4^–0.05 s^−1^ and strain of 0.6 was 0.17–0.73, and the Q-value was 328.88–546.90 KJ/mol. The m increased at a very low strain rate, and *η* maximized at a low *Q*-value. 

(2) The dynamic deformation mechanism map (DMM) thermal processing showed the machine-shaping of Ti–6Al–4V as follows: 800–840 °C and strain rate at 0.0005–0.001 s^−1^; 860–875 °C and strain rate at 0.001–0.005 s^−1^; the rheological instability zone was at 800–840 °C and strain rate at 0.034–0.05 s^−1^. 

(3) The deformation mechanism of RWS Ti–6Al–4V involving dislocation count was clarified. The dislocation quantitative relationship of this alloy based on the modulus compensation stress and Burgers vector compensation was determined, and the low-temperature superplastic compression deformation mechanism was predicted. 

## Figures and Tables

**Figure 1 materials-12-03520-f001:**
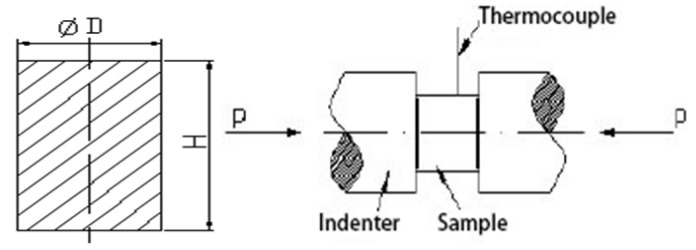
Schematic diagram of compression tests.

**Figure 2 materials-12-03520-f002:**
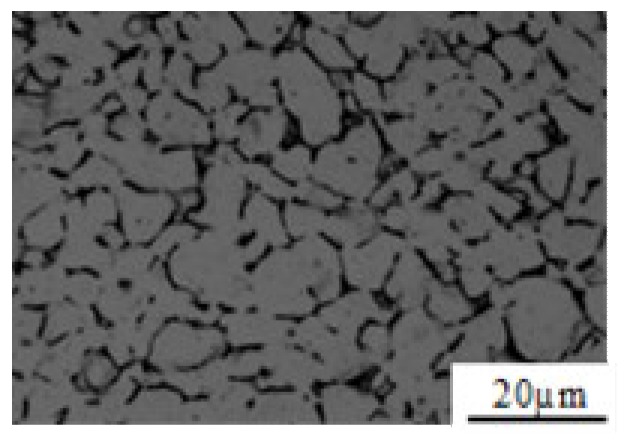
Optical micrograph of Ti–6Al–4V prior to deformation.

**Figure 3 materials-12-03520-f003:**
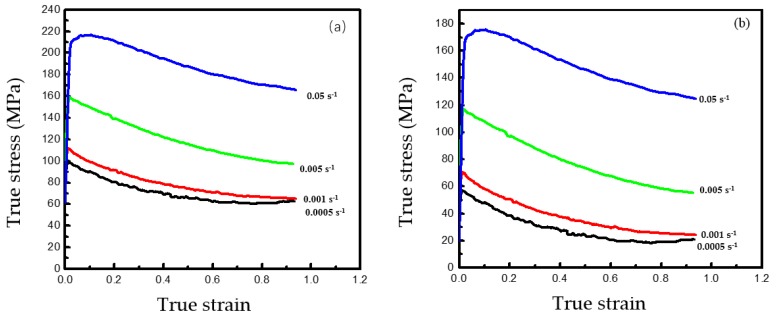
True stress–strain curves for the Ti–6Al–4V alloy under the tested conditions: T = (**a**) 800 °C, (**b**) 830 °C, (**c**) 860 °C, and (**d**) 890 °C.

**Figure 4 materials-12-03520-f004:**
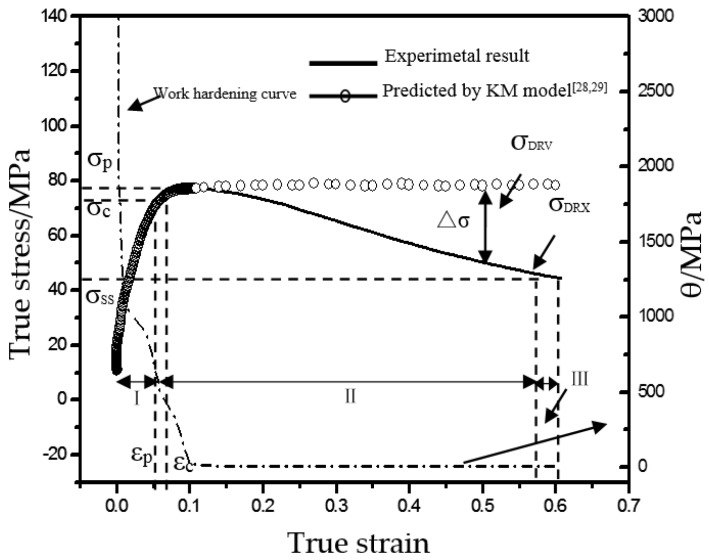
Flow stress curve and working hardening rate curve of Ti–6Al–4V.

**Figure 5 materials-12-03520-f005:**
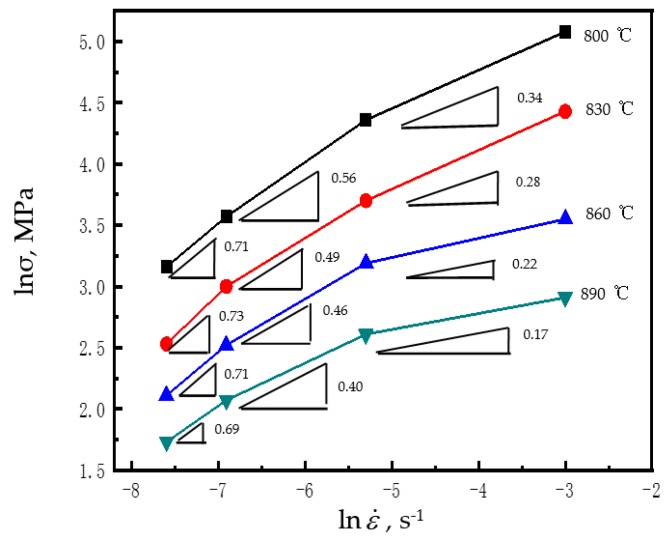
Strain rate sensitivity (m) of the superplastic flow.

**Figure 6 materials-12-03520-f006:**
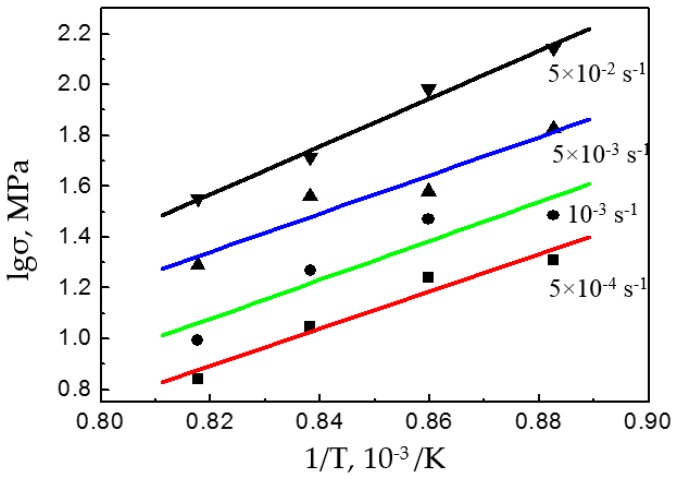
lgσ-1/T curve of the Ti–6Al–4V alloy.

**Figure 7 materials-12-03520-f007:**
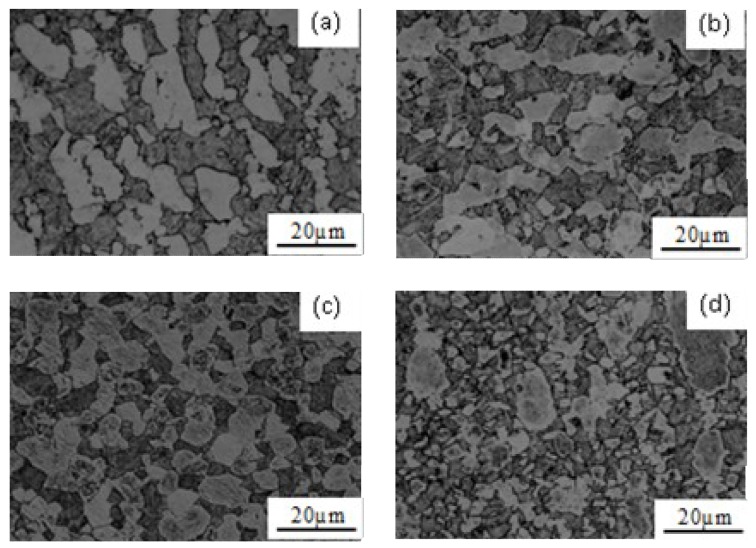
Microstructures of the TC4 alloy deformed at different strain rates (T = 860 °C). (**a**) 5 × 10^−4^ s^−1^; (**b**) 10^−3^ s^−1^; (**c**) 5 × 10^−3^ s^−1^; (**d**) 5 × 10^−2^ s^−1.^

**Figure 8 materials-12-03520-f008:**
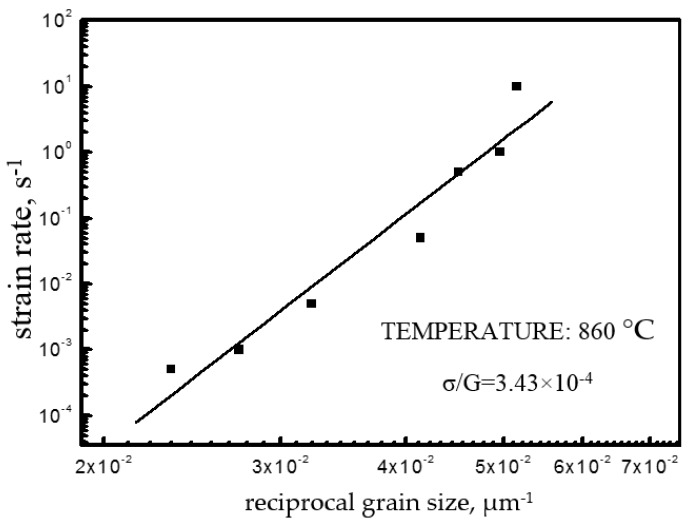
The variation in strain rate as a function of reciprocal grain size in Ti–6Al–4V.

**Figure 9 materials-12-03520-f009:**
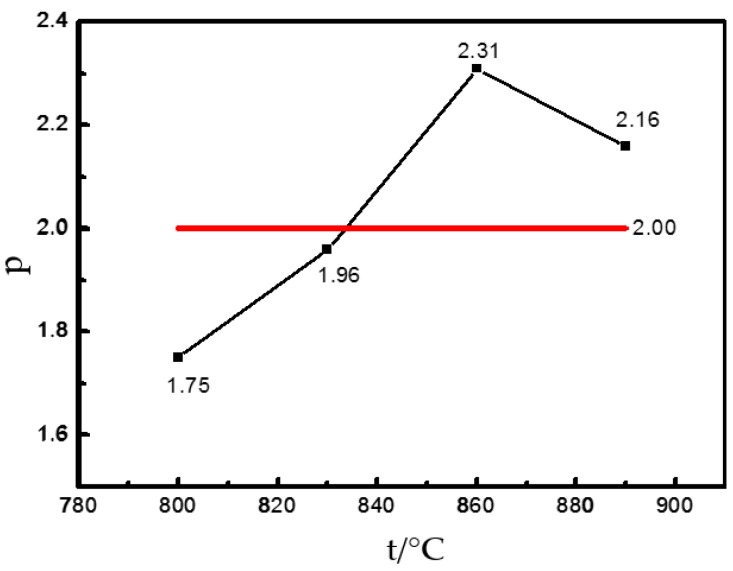
Grain size vs. temperature curve of the GH4742 superalloy.

**Figure 10 materials-12-03520-f010:**
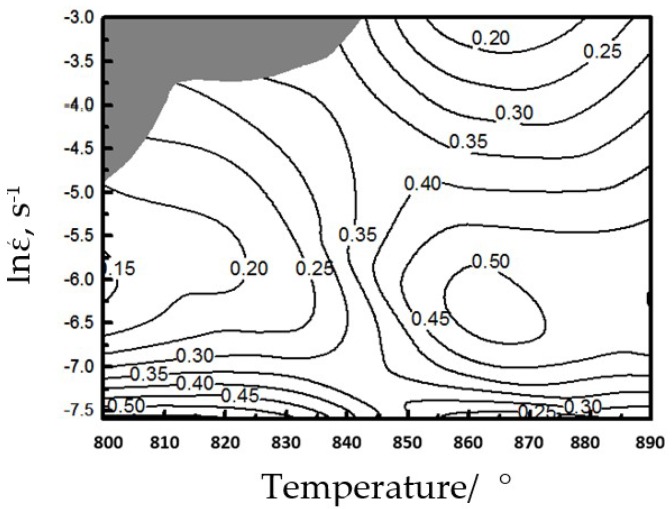
Processing maps for the Ti–6Al–4V alloy.

**Figure 11 materials-12-03520-f011:**
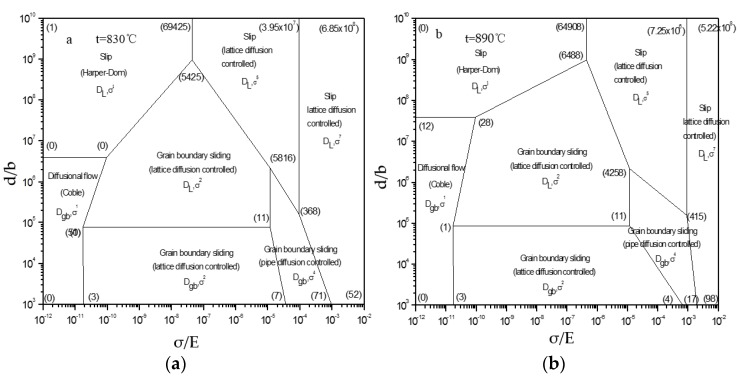
Deformation mechanism maps for Ti alloy at (**a**) 830 °C and (**b**) 890 °C.

**Figure 12 materials-12-03520-f012:**
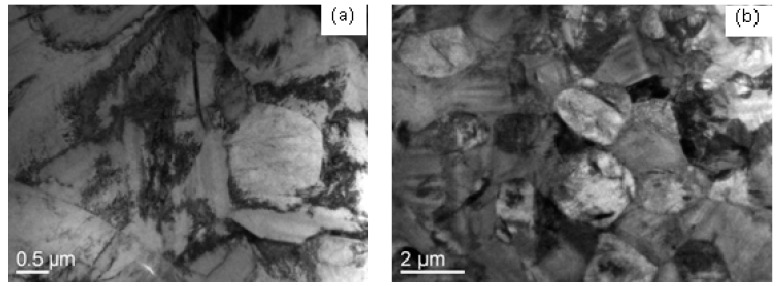
TEM microstructures of the Ti–6Al–4V alloy at 890 °C (ε = 0.6). (**a**) 5 × 10^−^^4^ s^−1^ and (**b**) 0.05 s^−1^.

**Table 1 materials-12-03520-t001:** Strain range of softening mechanism of Ti–6Al–4V at 860 °C and different strain rates.

$\dot{\varepsilon }$/s^−1^	5 × 10^−4^	10^−3^	5 × 10^−3^	5 × 10^−2^
**ε**	0.11–0.31	0.10–0.38	0.09–0.46	0.14–0.57

**Table 2 materials-12-03520-t002:** Activation energy Q (±7.7) during superplastic deformation of the TC4 alloy.

	T, °C	800	830	860	890
Strain Rate $\dot{\varepsilon }$, s^−1^					
5 × 10^−4^		328.88	400.89	429.08	389.35
5 × 10^−3^		344.52	418.80	449.33	407.99
10^−3^		338.96	412.09	442.14	401.44
5 × 10^−2^		427.54	519.22	546.90	505.87

**Table 3 materials-12-03520-t003:** Activation energies of self-diffusion and superplastic deformation of titanium alloys.

Alloy	Temperature Range T, °C	Activation Energy *Q*, J/mol	Ref.
Ti–6Al–4V	800–950	189	Arieli, Rosen [33]
Ti–6Al–4V	850–910	189–416	Mackey
Ti–6Al–4V	815–927	189–218	Wert, Paton
Self-diffusion, phase α		169	Dyment [34]
Self-diffusion, phase β		131	Pontau, Lazarus [35]

**Table 4 materials-12-03520-t004:** Physical parameters of titanium alloy [41].

b = 2.95 × 10^−10^ m	E = 2.87 × 10^5^ MPa	ν = 0.34	k = 1.38 × 10^−23^ J/K
D_L830 °C_ = 8.01 × 10^−16^ m^2^·s^−1^		D_L830 °C_ = 3.75 × 10^−16^ m^2^·s^−1^	
D_gb890 °C_ = 8.5 × 10^−11^ m^2^·s^−1^		D_gb890 °C_ = 6.3 × 10^−10^ m^2^·s^−1^	

**Table 5 materials-12-03520-t005:** Calculated results for hot compression deformation of the Ti–6Al–4V alloy.

T/°C	(*d/b*) × 10^−7^	(*σ/E*) × 10^4^	*έ*/(10^−4^·s^−1^)
830	8.2–11.7	8.2–42.9	5–100
890	7.7–9.3	2.4–21.5	5–100

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
