# Peer review of "Rheological Law and Mechanism for Superplastic Deformation of Ti–6Al–4V"

_materials, 2019, doi:10.3390/ma12213520_

Round 1

Reviewer 1 Report

The authors have obtained interesting results about deformation behaviour of Ti alloy with high deformation rate. However the results have limited application because the real superplastic deformation occurs not only at compression condition, but mainly at tension.

The authors used only a few recent references. In order to enhance the introduction and analysis of the results, I would recommend the authors several recent studies on the mechanisms of deformation under superplasticity conditions.

https://doi.org/10.1016/j.jmapro.2019.06.033

https://doi.org/10.3390/ma12111756

https://doi.org/10.3103/S1067821218060032

Did you take into account the microstructure evolution during the superplastic deformation? Usually microstructure changes have a strong effect on mechanisms of superplastic deformation.

Reviewer 2 Report

The submitted paper considers the investigation of the deformation mechanisms during low-temperature superplastic deformation of Ti-6Al-4V alloy. Based on the experimental data, the activation energy of thermal deformation and parameters for hot deformation were calculated. The paper is well written and is recommended to be printed.

 Some issues:

Line 71, How many samples were used per condition?

Line 90, please add standard deviation of measurements.

Table 2, please add errors to data

Figure 7, please replace by graph with better quality (x-scale is not visible)

Reviewer 3 Report

General comments:

The derivation of deformation maps is important. This work seems to achieve that, which is good.

I do not think 800C qualifies as "low-temperature". Even if it is below the maximum service temperature, it is still very hot.  I think another phrase should be used.  Ti-6Al-4V is used at room temperature, too.

More explanation is required for how equations were derived or what sources were used.

Page 1

Line 9: it is not correct English to say "behaviors and mechanisms of Ti-6Al-4V...". You may, however, say "the behaviors of and mechanisms acting in Ti-6Al-4V...". Line 26-27: please provide a source of information to support your claim that Ti-6Al-4V is the most widely-used titanium alloy. Line 41: remove the final wor din the line ("the") Line 38: "the alloy with low formability rather hardly forms a complicated shape" is not correct English, please change it, for example: "alloys with low formability can not easily be manufactured into a component with a complicated shape". Line 44: "which are related with hardening" is not specific. I think the mechanisms actually lead to softening, not hardening, but are driven by a high dislocation density rather than by "hardening". Please can you rewrite this sentence to be clearer. Line 49: "to Al alloys" should be "for Al alloys".

Page 2

Lines 79-85: Please make sure chemical formulae are written in accordance with proper formatting rules (capital and lower-case letters, subscript numbers where needed, etc.) Lines 82-84: please describe the operating conditions, and detectors used on TEM. Please decribe what an Olympus M3 ImageMeter does? Line 90: Please provide uncertainty estimates in the measurements (e.g. (54+/-2)%, (11.5+/-0.1)µm).

Page 3
Lines 94-98: Please ensure you are not causing the image to overlap the caption. This may not be your fault, but please check that it is not your actions that cause this. Lines 101-103: "Basically, the computed flow... closer to the real ones". These sentences do not make sense. To what computed stress-strain curves are you referring? THis is the first time you have written about computing stress-strain curves. Line 107: there is no figure 3e. Line 106-107: I odo not agree with this interpretation of figures 3c and 3d. There seems to be an upper yield stress and sudden drop when the plastic deformation begins. The linear "initial rise" is elastic deformation, not flow stress. The flow stress begins when plastic deformation begins. In figures 3c and 3d, there is an immediate drop in flow stress. Line 107-109: Do you have direct evidence for this, or is that just taken from standard theory of deformation? If you have evidence for these mechanisms, please provide it. If it is a deduction from theory, please make this clear.

Page 4 Line 136: please use the correct symbol for partial differential, not an upside-down "e". Line 137-138: it is not necessary to describe the behaviour - this is what the plot is for. Line 138: please introduce the KM model for the readers - this is the first time it is written and no citations or explanations of it are given. Line 160: please move the label foe critical strain for dynamic recrystallisation - it overlaps the line in figure 4 and is difficult to see. Line 143: Please explain why you define zone III. I cannot see any stabilisation. I see that the line becomes slightly horizontal, but it is still decreasing quickly. What defines the change from zone II to zone III? Line 169-170: I do not understand this point. You seem to be comparing the dependence of m on strain rate with the amount of softening with absolute strain? I cannot see how they can be compared. Please try rewriting this to make it clearer, or explaining it more thoroughly to help readers.

Page 5

Line 173: The units of stress are currently written as "MMPa". I think that is incorrect. Line 191: "K" does not appear in the equation, "k" does. Please correct this. Line 194: I think you mean "lg[sigma] vs. 1/T was plotted and has a slope of...", that is a more correct way to phrase this sentence.

Page 6

Line 207-212: Please consider the precision of your results carefully. Can you justify giving values to five significant figures? Please provide an uncertainty estimate for all measurements and calculated values. Also, it is not necessary to repeat values from the table, simply refer to the table. Line 214-216: Have you got evidence to support this, or is it what you conclude from your activation measurements? Please give some more details and cite a source if you have read this in literature. Line 220-223: Do you know the power law applies to your conditions and material? Please provide literature evidence for this. Line 231: Is m, the "strain rate sensitivity", that is mentioned here the same "strain rate sensitivity" as in figure 5? If so, please make this clear. The text you have written makes it look like it is a new quantity in the paper.

Page 7

Line 228-233: Please provide some more detail about how this is derived. As the derivation stands, it seems like you have invented the derivation, so please provide some more explanation. If this is taken only from source 32, please say this and then you do not need to reproduce the derivation in detail. Line 239: please explain where equation 12 comes from. The zeta symbol is not used anywhere else in the paper. Line 242: do you mean shaded area? Line 244: something strange has happened with the labels. Please check and fix this. Line 247-252: please provide more explanation for these numbers and not just list them.

Page 8

Line 255: what is the RMS mechanism? Line 259-260: Make sure you do not use any symbols that you haven't used previously. Line 266: where is this plot, or the results of your analysis for crystal sizes? Line 272-275: "normalization" is the correct word here, not "compensation". Please change these two instances.

Page 9

Line 291-296: Please can you label the regions on the maps, instead of referring to their vertices. This would be much easier for readers to understand. Line 300-306: I think it would be good if you could provide some examples of the values you can calculate or derive for the quantities you write about here. Line 306-307: Please explain how figure 9a shows dislocation motion you describe. Line 307-311: Please explain how figure 9b shows the effects you describe.

Page 10

Line 355: what is DMM? You haven't defined this abbreviation.

Round 2

Reviewer 3 Report

Thank you for addressing my comments and for the explanations you have provided.  I am satisfied by most of your changes, but there are still a small number on which I would like further clarification:

Page 3, line 109: The upper yield stress phenomenon is only seen in figures 3c and 3d.  There is no such initial drop in yield stress in figures 3a and 3b.  The authors should explain this. Page 4, line 146: I still do not understand why you define zone III.  I do not see anything in your results that you could describe as a stable state.  If you have such data, please explain this in the text.  If this definition comes from some prior theory, please explain this in the text. Page 5, lines 193-196: I recommend you use "ln" for natural logarithm, rather than "lg".  You are not incorrect and in this paper, it is obvious that your take natural logs, but I believe "lg" can lead to confusion with decadic (base 10) logarithms ("log") in other papers, so I discourage the use of "lg" at any time.  However, you may leave "lg" if you wish as it is technically correct.
